# A FAST KERNEL-BASED CONDITIONAL INDEPENDENCE TEST WITH APPLICATION TO CAUSAL DISCOVERY

## ABSTRACT

Kernel-based conditional independence (KCI) testing is a powerful nonparametric method commonly employed in causal discovery tasks. Despite its flexibility and statistical reliability, cubic computational complexity limits its application to large datasets. To address this computational bottleneck, we propose *FastKCI*, a scalable and parallelizable kernel-based conditional independence test that utilizes a mixture-of-experts approach inspired by embarrassingly parallel inference techniques for Gaussian processes. By partitioning the dataset based on a Gaussian mixture model over the conditioning variables, FastKCI conducts local KCI tests in parallel, aggregating the results using an importance-weighted sampling scheme. Experiments on synthetic datasets and benchmarks on real-world production data validate that FastKCI maintains the statistical power of the original KCI test while achieving substantial computational speedups. FastKCI thus represents a practical and efficient solution for conditional independence testing in causal inference on large-scale data.

## 1 INTRODUCTION

Conditional independence (CI) testing is a fundamental operation in causal discovery and structure learning. Widely used algorithms such as the PC algorithm (Spirtes & Glymour, 1991) and Fast Causal Inference (Spirtes, 2001) rely on CI tests to recover the causal skeleton of a graph from observational data. The core statistical question is whether two variables $X$ and $Y$ are independent given a conditioning set $Z$, that is, whether $X \perp\!\!\!\perp Y \mid Z$. Despite being frequently adapted to fields like neuroscience (Smith et al., 2011), climate research (Ebert-Uphoff & Deng, 2012) or economics (Awokuse & Bessler, 2003), CI testing remains the computational bottleneck in constraint-based causal discovery, especially as sample size increases (Agarwal et al., 2023; Le et al., 2019; Shiragur et al., 2024).

Standard CI tests suit different types of data and assumptions: Traditional tests like the Fisher-$Z$ test (Fisher, 1921) assume linear Gaussian data, while discrete tests such as $\chi^2$ require categorical variables. More recent approaches avoid strong distributional assumptions by leveraging nonparametric techniques such as kernel methods. Among these, the Kernel-based Conditional Independence test (KCI) (Zhang et al., 2012) has become a standard choice due to its flexibility and empirical power. KCI is based on Hilbert space embeddings and computes dependence via kernel covariance operators, making it applicable to arbitrary continuous distributions.

KCI requires operations on $n \times n$ Gram matrices and matrix inversions, resulting in $\mathcal{O}(n^3)$ runtime per test, which makes it infeasible for large-scale applications. Recent work has sought to mitigate this cost via sample splitting (Pogodin et al., 2024), random Fourier features (Strobl et al., 2018), neural network approximations (Doran et al., 2014), and randomization-based tests (Shah & Peters, 2020). However, these approximations can degrade statistical power and require additional tuning.

Our goal is to accelerate the KCI test without sacrificing its statistical rigor and nonparametric flexibility. To this end, we propose *FastKCI*, a novel variant that leverages ideas from embarrassingly parallel inference in Gaussian processes (Zhang & Williamson, 2020). FastKCI partitions data based on a generative model in the conditioning set $Z$, performs KCI tests in parallel on each partition, and aggregates the test statistics using an importance weighting scheme. This blockwise strategy enables significant computational speedups, especially in multi-core or distributed environments.

**Our contributions** are therefore a scalable and parallelizable conditional independence test, FastKCI, that significantly accelerates KCI by using a novel partition-based strategy combined with importance weighting. We experimentally demonstrate that FastKCI retains the statistical performance of KCI while achieving runtime improvements across synthetic and real-world datasets.

## 2 RELATED WORK

Due to the growing importance of causal inference, considerable research has been devoted to discovering efficient methods to identify causal structures from observational data (Zanga et al., 2022). Traditional causal discovery methods are often categorized as constraint-based or score-based. Constraint-based methods, such as the PC algorithm (Spirtes & Glymour, 1991) or FCI (Spirtes et al., 1995), rely on CI tests to identify the underlying causal structure. Score-based methods, such as Greedy Equivalence Search (Chickering, 2002), evaluate causal structures based on scoring criteria. Both approaches face computational challenges: constraint-based methods due to intensive CI testing, and score-based methods due to an exponentially large search space.

Existing work has improved the efficiency of the PC algorithm by reducing unnecessary CI tests (Steck & Tresp, 1999), optimizing the overall search procedure itself—through order-independent execution (Colombo & Maathuis, 2014), skipping costly orientation steps (Colombo et al., 2012), sparsity-aware pruning (Kalisch & Bühlmann, 2007), divide-and-conquer partitioning (Huang & Zhou, 2022), and parallelism of the CI tests (Le et al., 2019; Zarebavani et al., 2020; Hagedorn et al., 2022). These methods, however, do not address the cubic-time bottleneck of kernel-based CI tests.

Attempts in the literature to address this issue take several forms. The original KCI paper already derived an analytic $\Gamma$-approximation of the null distribution and proposed simple median-heuristic bandwidth choices to avoid costly resampling (Zhang et al., 2012). Strobl et al. (2018) accelerate KCIT by replacing the full kernel matrices with an $m$-dimensional random Fourier-feature approximation. Doran et al. (2014) re-express conditional independence as a single kernel two-sample problem by restricting permutations of $(X, Y)$, thereby changing the test statistic while lowering runtime. Zhang et al. (2022) eliminate kernel eigen-decompositions altogether by regressing $X$ and $Y$ on $Z$ and measuring residual similarity with a lightweight kernel. Additional ideas include calibrating test statistics with locality-based permutations in the conditioning set (Kim et al., 2022), evaluating analytic kernel embeddings at a finite set of landmark points (Scetbon et al., 2021), and controlling small-sample bias via data splitting (Pogodin et al., 2024). However, these approaches can compromise statistical power, particularly under complex nonlinear dependencies and in high-dimensional conditioning sets. Concurrent with our work, Guan & Kuang (2025) proposed an Ensemble Conditional Independence Test, which independently adopts a similar divide-and-aggregate strategy by partitioning the data, applying a generic base conditional independence test to each subset, and combining the resulting p-values via stable-distribution-based aggregation. While conceptually related, their approach and ours differ substantially in both the sample partitioning scheme and the aggregation mechanism.

Our method preserves the KCI statistic by evaluating it on Gaussian-mixture strata and aggregating with importance weights. We therefore leverage techniques from Gaussian Process regression, which often faces the identical problem of poor scalability due to cubic complexity[1] (Zhang & Williamson, 2020). A common solution is to assume the underlying distribution of the covariates $Z = \{z_1, \ldots, z_j\}$ to be a mixture-of-experts (MoE) (Jordan & Jacobs, 1994). Local approaches, as in Gramacy & Lee (2008), then aggregate over multiple partitions by MCMC. Zhang & Williamson (2020) propose an importance sampling approach to MoE that efficiently aggregates over multiple partitions. To the best of our knowledge, our approach – using importance-sampled partitions of the data, performing parallel kernel tests, and aggregating via importance weighting – represents the first attempt explicitly bridging embarrassingly parallel inference with scalable kernel-based CI testing.

## 3 BACKGROUND

The KCI builds on a notion of conditional independence introduced by Fukumizu et al. (2007). Let $(\Omega, \mathcal{F}, \mathbb{P})$ be a probability space and $X \in \mathcal{X}, Y \in \mathcal{Y}, Z \in \mathcal{Z}$ random variables. For each domain we

---

[1]In fact, as KCI solves GP regression problems to find the RKHS bandwidth, it is a sub-problem of CI.

fix a measurable, bounded and characteristic kernel, e.g. the Gaussian RBF, denoted $k_X, k_Y, k_Z$, with corresponding reproducing-kernel Hilbert spaces (RKHS) $\mathcal{H}_X, \mathcal{H}_Y, \mathcal{H}_Z$. Feature maps are denoted $\varphi_X(x) = k_X(x, \cdot)$ etc. Expectations are shorthand $\mathbb{E}[\cdot]$, tensor products $\otimes$, and centered features $\tilde{\varphi}_X := \varphi_X - \mu_X$ where $\mu_X = \mathbb{E}[\varphi_X(X)]$.

### 3.1 COVARIANCE AND CONDITIONAL COVARIANCE OPERATORS

The *cross-covariance operator* $\Sigma_{XY} : \mathcal{H}_X \to \mathcal{H}_Y$ is the bounded linear map satisfying

$$\langle g, \Sigma_{XY} f \rangle = \mathbb{E}\big[\langle \tilde{\varphi}_X(X), f \rangle_X \, \langle \tilde{\varphi}_Y(Y), g \rangle_Y \big] \quad \forall f \in \mathcal{H}_X, \, g \in \mathcal{H}_Y.$$

An analogous definition yields $\Sigma_{XZ}, \Sigma_{ZZ}$. Provided $\Sigma_{ZZ}$ is injective[2], conditional covariance is

$$\Sigma_{XY|Z} := \Sigma_{XY} - \Sigma_{XZ} \Sigma_{ZZ}^{-1} \Sigma_{ZY}.$$

**Proposition 1** (Fukumizu et al., 2007). *With characteristic kernels,* $X \perp\!\!\!\perp Y \mid Z \iff \Sigma_{XY|Z} = 0$.

Hence testing conditional independence reduces to checking whether this operator is null.

### 3.2 FINITE-SAMPLE KCI STATISTIC

Given $n$ observations $\{(x_i, y_i, z_i)\}_{i=1}^n$, assemble Gram matrices $K_X, K_Y, K_Z \in \mathbb{R}^{n \times n}$ with $(K_X)_{ij} = k_X(x_i, x_j)$, $K_Y, K_Z$ analogously. Let $H := I_n - \frac{1}{n}\mathbf{1}\mathbf{1}^\top$ and define the projection onto $Z$–residuals

$$R_Z := I_n - K_Z\big(K_Z + \lambda I_n\big)^{-1}, \quad \lambda > 0. \tag{1}$$

Intuitively, $R_Z$ acts like a residual operator that removes components explained by Z, $R_z f$ is approximately the part of $f$ orthogonal to functions of $Z$. Residualized, centered kernels are $\tilde{K}_X = R_Z H K_X H R_Z$ and $\tilde{K}_Y$ analogously.

The KCI test statistic of Zhang et al. (2012) is then given by the Hilbert–Schmidt norm of the cross-covariance between residuals:

$$T_{\mathrm{KCI}} := \frac{1}{n} \mathrm{Tr}(\tilde{K}_X \tilde{K}_Y). \tag{2}$$

Zhang et al. (2012) show that under $H_0$ this statistic converges in distribution to a weighted sum of $\chi^2$ variables. In practice one uses a finite-sample null distribution to assess significance. We follow Proposition 5 in Zhang et al. (2012), using a spectral approach to simulate the null: We compute the eigenvalues $\lambda_m$ of a normalized covariance operator associated with $\tilde{K}_X$ and $\tilde{K}_Y$, then generate null samples and set

$$T_{\mathrm{null}}^{(b)} = \sum_{m=1}^n \lambda_m \chi^2_{1,m,b}, \qquad b = 1, \ldots, B, \tag{3}$$

with i.i.d. $\chi^2_1$ variates $\chi^2_{1,m,b}$. This approximate distribution of $T_{\mathrm{KCI}}$ under $H_0$ obtains a $p$-value. The full KCI procedure is given in Algorithm 2.

### 3.3 COMPUTATIONAL COMPLEXITY OF THE KCI

A key drawback of KCI is its heavy computation for large $n$. Constructing and manipulating $n \times n$ Gram matrices is $\mathcal{O}(n^2)$ in memory. More critically, forming $\tilde{K}_X$ and $\tilde{K}_Y$ requires solving a linear system or eigen-decomposition on $K_Z \in \mathbb{R}^{n \times n}$. The inversion $(K_Z + \varepsilon I)^{-1}$ costs $\mathcal{O}(n^3)$ time in general. Generating the null distribution via eigenvalues also incurs an $\mathcal{O}(n^3)$ decomposition of the $n \times n$ matrix $U = (I - R_Z)K_X(I - R_Z)$ and related matrices. Thus, the overall complexity of KCI scales cubically in the sample size $n$. This cubic bottleneck severely limits the test's applicability to large datasets, particularly when it must be repeated many times as in constraint-based causal discovery.

---

[2]With a characteristic kernel this holds on the closure of its range.

## 4 FASTKCI: A SCALABLE AND PARALLEL KERNEL-BASED CONDITIONAL INDEPENDENCE TEST

To overcome the $\mathcal{O}(n^3)$ bottleneck, we propose *FastKCI*, which leverages a mixture-of-experts model and importance sampling. The core idea is to break the full kernel computation into $V$ smaller pieces (*experts*) corresponding to data partitions, compute local CI statistics on each piece, and then recombine them to recover the global statistic. By doing so, FastKCI achieves significant speed-ups – roughly on the order of $1/V^2$ of the cost of KCI – while maintaining the test's correctness under mild assumptions.

We assume that the distribution of the conditioning variable $Z$ can be approximated by a mixture of $V$ Gaussian components.

**Assumption 1.** *Let $U$ be a latent cluster assignment variable taking values in $\{1, \ldots, V\}$. For each $i$, we posit a model:*

$$U_i \sim \text{Categorical}(\pi_1, \ldots, \pi_V), \quad \pi \sim \text{Dirichlet}(\alpha), \tag{4}$$

$$z_i \mid (U_i = v) \sim \mathcal{N}(\mu_v^Z, \Sigma_v^Z), \quad (\mu_v^Z, \Sigma_v^Z) \sim \text{Normal-InvWishart}(\mu_0, \lambda_0, \Psi, \nu). \tag{5}$$

Thus, $z_i$ are i.i.d. draws from a $V$-component Gaussian mixture with unknown means $\mu_v^Z$ and covariances $\Sigma_v^Z$. We place a weak Normal-Inverse-Wishart prior on these parameters to allow uncertainty. This mixture-of-experts prior on $Z$ guides partitioning of the dataset. Intuitively, if $Z$ has a multimodal or complex distribution, this model lets us divide the data into $V$ components $\mathcal{C}_1, \ldots, \mathcal{C}_V$ (with $\mathcal{C}_v = \{i \mid U_i = v\}$) such that within each $\mathcal{C}_v$, $Z$ is roughly Gaussian. We do not assume anything restrictive about $X$ and $Y$ globally. By conditioning on $\mathcal{C}_v$, the relationship between $X$ and $Y$ can be analyzed locally. In particular, conditioned on a given partition $U_{1:n} = (U_1, \ldots, U_n)$, the kernel matrices $K_X, K_Y, K_Z$ acquire an approximate block structure: after permuting indices by cluster, each matrix breaks into $V$ blocks (sub-matrices) corresponding to points in the same cluster, with negligible entries for cross-cluster pairs, especially if clusters are well-separated in $Z$. Each component $v$ defines a local sub-problem of size $n_v = |\mathcal{C}_v|$, and within that component we can perform a CI test on the restricted data $\{(x_i, y_i, z_i) : i \in \mathcal{C}_v\}$. This yields a local test statistic $T_v$ and local null distribution for component $v$. By appropriately combining these local results, we can recover a valid global test statistic without ever computing the full $n \times n$ kernel on all data at once.

The choice of a Gaussian Mixture Model assumption is motivated by its power as a universal approximator of densities, capable of modeling a wide variety of complex, multi-modal, and non-Gaussian distributions with high fidelity (McLachlan & Peel, 2000). The goal of FastKCI is not to assume the data is strictly Gaussian, but to leverage a flexible framework to create sensible, localized partitions of the conditioning set. Importantly, our empirical results in Section 5 (and Appendix A.1) demonstrate that FastKCI exhibits robust performance even when this assumption is misspecified or the true underlying distribution is unknown.

In the next paragraphs, we explain the procedure in detail.

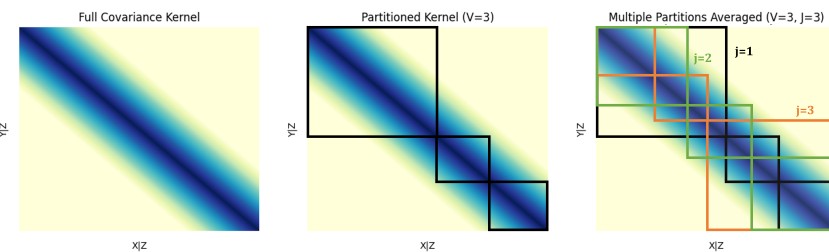

Figure 1: Motivation of the partitioning scheme in the data. The full covariance kernel estimation (left) is inefficient, while partitioning the data into components a single time (middle) may neglect some of the covariance structure. We propose to use multiple partitions $J$ (right) in parallel. We combine them using importance sampling. The figure is inspired by Zhang (2020).

**Partition Sampling.** As visible in Figure 1, for partition sampling rounds $j = 1, \ldots, J$ we independently draw a cluster assignment $U_{1:n}^{(j)} = (U_1^{(j)}, \ldots, U_n^{(j)})$ according to the MoE model on $Z$.

The mixture prior is fit to the empirical $Z$ data. In practice, we use a lightweight empirical-Bayes partitioning scheme (see Appendix B) that samples mixture components around the empirical mean of $Z$ instead of performing full EM or full posterior inference. For example, one partition sample $j$ samples mixture parameters $(\mu_1^Z, \ldots, \mu_V^Z, \Sigma_1^Z, \ldots, \Sigma_V^Z) \sim P(\mu, \Sigma)$ from the NIW prior and mixture weights $\pi$, then assigns each point $i$ to a component $U_i^{(j)} = v$ with probability

$$P\left(U_i^{(j)} = v \mid z_i\right) \propto \pi_v \mathcal{N}\left(z_i \mid \mu_v^Z, \Sigma_v^Z\right).$$

Each such draw yields a partition $\{\mathcal{C}_1^{(j)}, \ldots, \mathcal{C}_V^{(j)}\}$ of $\{1, \ldots, n\}$.

**Local RKHS Embedding, Test Statistic and Null Distribution.** For each partition $j$ and for each cluster $v \in \{1, \ldots, V\}$, let $\mathcal{C}_v^{(j)} = \{i \mid U_i^{(j)} = v\}$ be indices in the cluster of size $n_v^{(j)} = |\mathcal{C}_v^{(j)}|$. We form $n_v^{(j)} \times n_v^{(j)}$ Gram matrices $K_X^{(j,v)}$, $K_Y^{(j,v)}$, and $K_Z^{(j,v)}$. $\tilde{K}_X^{(j,v)}$ and $\tilde{K}_Y^{(j,v)}$ are calculated analogously to Equation 1 with local regression operators. The local test statistic follows as

$$T_v^{(j)} = \frac{1}{n_v^{(j)}} \mathrm{Tr}\left(\tilde{K}_X^{(j,v)} \tilde{K}_Y^{(j,v)}\right).$$

We generate a set of $B$ null samples $\{T_{v,\mathrm{null}}^{(j,b)}\}$ for each cluster by applying the spectral method in Equation 3 block-wise. Finally, we record a log-likelihood score for each cluster: let $\mathcal{L}\left(X^{(j,v)}\right) = \log P\left(X\left(\mathcal{C}_v^{(j)}\right) \mid Z\left(\mathcal{C}_v^{(j)}\right)\right)$ and $\mathcal{L}\left(Y^{(j,v)}\right)$ analogously be the log marginal likelihoods of the $X$ and $Y$ data in cluster $v$ given $Z$. We use this measure to calculate a likelihood $\ell^{(j,v)} \propto P(X^{(j)}, Y^{(j)} \mid U^{(j)})$.

**Aggregation over $V$.** We aggregate the partition-wide test statistic as the sum of the cluster statistics.

$$T^{(j)} = \sum_v T_v^{(j)}$$

This recovers the full trace of the product of block-wise diagonal $\tilde{K}_x^{(j)} \tilde{K}_y^{(j)}$. Since under $H_0$ the cluster test statistics are approximately independent (different clusters involve disjoint data) and each follows a weighted $\chi^2$ distribution, their sum $T_{\mathrm{null}}^{(j,b)}$ is a valid sample from the null distribution for the whole partition $j$. Thus, we also aggregate each null sample into the sum.

$$T_{\mathrm{null}}^{(j,b)} = \sum_v T_{v,\mathrm{null}}^{(j,b)}$$

**Aggregation over $J$.** We apply a softmax to the log-likelihoods to obtain per-partition importance weights

$$w_j = \frac{\exp\left(\sum_v \ell^{(j,v)}\right)}{\sum_j \exp\left(\sum_v \ell^{(j,v)}\right)}$$

The final test statistic is a weighted average across all $J$ partitions:

$$T_{\mathrm{FastKCI}} = \sum_{j=1}^J w_j T^{(j)}, \quad \text{where } w_j \propto P(X^{(j)}, Y^{(j)} \mid U^{(j)})$$

and the combined null distribution is taken as the mixture of all partition null samples with the same weights. In practice, we merge the $J$ sets of null samples $\{T_{\mathrm{null}}^{(j,b)}\}$ into one weighted empirical distribution. Specifically, we compute the weighted empirical cumulative density $F_{T,\mathrm{null}}(t) = \sum_{j=1}^J w_j \left(\frac{1}{B} \sum_{b=1}^B \mathbf{1}\{T_{\mathrm{null}}^{(j,b)} \le t\}\right)$, which is a mixture of the $J$ null distributions. See also Appendix B for implementation details.

## 4.1 THEORETICAL INSIGHT

Under the null hypothesis $H_0 : X \perp\!\!\!\perp Y \mid Z$, each block statistic $T_v^{(j)}$ is, by exactly the same argument as in Zhang et al. (2012), asymptotically a weighted sum of independent $\chi_1^2$ variables,

$$T_v^{(j)} \xrightarrow{d} \sum_m \lambda_{j,v,m}\, \chi_{1,(j,v,m)}^2,$$

with non-negative weights $\lambda_{j,v,m}$ determined by the eigenvalues of the blockwise covariance operators.

Different clusters use disjoint subsets of the data, hence their statistics are (asymptotically) independent; the sum $T^{(j)} = \sum_v T_v^{(j)}$ is therefore a new weighted $\chi^2$ mixture whose weights are a union of $\lambda_{j,v,m}$. The convex combination of these weighted $\chi^2$ mixtures is itself a weighted $\chi^2$ mixture with the same weights $w_j$ :

$$T_{\text{FastKCI}} \xrightarrow{d} \sum_{j=1}^{J} \sum_{v=1}^{V} \sum_m w_j\, \lambda_{j,v,m}\, \chi_{1,(j,v,m)}^2.$$

Hence, our method inherits exactly the same null-law template as classical KCI: a positive, finite linear combination of $\chi_1^2$ variables.[3] Consequently, under Assumption 1 and with $J \to \infty$, the test exhibits the same appealing statistical properties as the conventional KCI.

## 4.2 COMPUTATIONAL COMPLEXITY

Assuming balanced partitions, each block contains roughly $n/V$ samples. The complexity of KCI per block is then $\mathcal{O}((n/V)^3)$, and the total complexity becomes $\mathcal{O}(Jn^3/V^2)$. Since the $J$ partitions are fully parallelizable, the wall-clock cost is significantly reduced compared to the original $\mathcal{O}(n^3)$ cost.

While the formal complexity expression of FastKCI appears cubic in $n$, this perspective presumes that $V$ is a small, fixed constant. In practice, for the mixture model to effectively approximate the underlying distribution of the conditioning set $Z$, the number of components $V$ can increase with the sample size. This scaling ensures that the number of samples within each cluster $(n/V)$ remains manageable, preventing single partitions from becoming a computational bottleneck. If $V$ is chosen to scale with $n$, the effective computational complexity of FastKCI becomes nearly linear in $n$. We support this by our experiments, where we show manageable computation times in large samples by growing $V$.

The complete procedure is summarized in Algorithm 1.

---

**Algorithm 1** FastKCI: Fast and Parallel Kernel-based CI Test

---

**Require:** Dataset $\{(x_i, y_i, z_i)\}_{i=1}^n$, number of components $V$, number of partition sampling rounds $J$.
1: **for** $j = 1$ to $J$ **in parallel do**
2:     Sample $V$-component partition $U^{(j)}$ from $p(U \mid Z)$.
3:     **for** each component $v = 1$ to $V$ **do**
4:         Compute residualized kernels $K_{X|Z}^{(v)}, K_{Y|Z}^{(v)}$.
5:         Compute test statistic $T_v^{(j)} = \frac{1}{n_v}\text{Tr}(K_{X|Z}^{(v)} K_{Y|Z}^{(v)})$.
6:     **end for**
7:     Aggregate: $T^{(j)} = \sum_v T_v^{(j)}$.
8:     Compute importance weight $w_j \propto P(X^{(j)}, Y^{(j)} \mid U^{(j)})$.
9: **end for**
10: Normalize weights: $w_j \leftarrow w_j / \sum_j w_j$.
11: Compute final test statistic and null distribution by weighted averaging.
12: **return** $p$-value for $H_0 : X \perp\!\!\!\perp Y \mid Z$.

---

[3]The formal requirements are the standard KCI assumptions (characteristic kernels, boundedness) plus every cluster size $|\mathcal{C}_v^{(j)}| \to \infty$ as $n \to \infty$.

### 4.3 Hyperparameter Selection

The practical application of FastKCI requires the specification of the hyperparameters $J$ and $V$, for which we provide some guidance. The number of partitions $J$ is primarily determined by the available computational resources. Since FastKCI is parallelized across $J$ partitions, each processed independently, increasing $J$ in line with the number of available CPU nodes does not substantially increase runtime. Moreover, because the importance sampling procedure favors partitions that more accurately model the data, a larger $J$ generally yields improved results through additional sampling rounds.

The selection of the number of clusters $V$ is comparatively more challenging. When prior knowledge exists regarding the number of Gaussian components underlying the data-generating process, such information provides a natural choice for $V$. In the more typical scenario where the distribution of the conditioning set $Z$ is unknown, selecting $V$ involves a trade-off between sample size and cluster size. Increasing the number of clusters improves the approximation of the conditional distribution of $Z$ while simultaneously reducing the average cluster size (approximately $n/V$). Smaller clusters facilitate more efficient eigendecompositions, thereby enhancing scalability. However, excessively small clusters can introduce instability. For an ablation study on $V$, see Appendix A.1[4]

## 5 Experiments

To empirically validate the main result in Algorithm 1, we extensively study the performance of *FastKCI* and compare it to the KCI implementation provided in the `causal-learn` package (Zheng et al., 2024). We consider different scenarios, focusing on coverage, power and causal discovery[5]. Please find additional results concerning an ablation on $V$, and non-Gaussian conditioning sets in Appendix A.1.

### 5.1 Type-I-Error Comparison

We generate $n = 1200$ samples of random variables $X$, $Y$ and $Z$, with $X$ and $Y$ being drawn independently conditioned on $Z$. In our scenario, we examine the type I error with growing confounding set size $D = \{1, \ldots, 5\}$, with all variables effecting both $X$ and $Y$ (comparable to "Case II" in Zhang et al. (2012)). For the $Z_i$, we consider multiple ground-truth distributions, as a mixture of $V_{\text{true}} = \{1, 3, 10\}$ Gaussians. $X$ and $Y$ are generated as post-nonlinear causal model $g\left(\sum_i f_i(Z_i) + \varepsilon\right)$ where $f$ and $g$ are random mixtures of linear, cubic and `tanh` functions and $\varepsilon$ is independent across $X$ and $Y$.

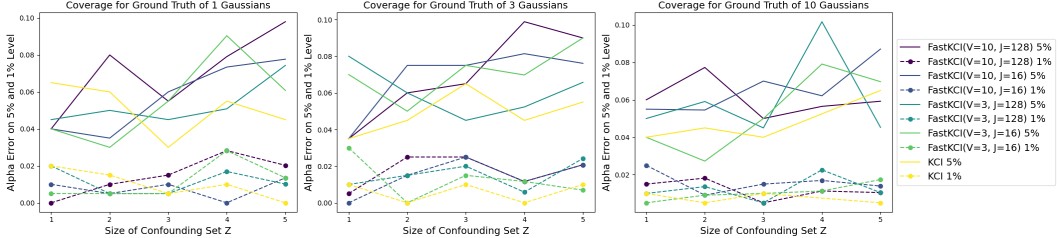

Figure 2: Simulated Type-I-Error ("Coverage") of the *FastKCI* and the KCI at 1% and 5% levels.

### 5.2 Power Comparison

We repeat the experiments from above, but make $X$ and $Y$ conditionally dependent by adding a small identical noise component $\nu \sim \mathcal{N}(0, \sigma_{\text{vio}}^2)$ to both random variables, in order to assess the type II

---

[4]More systematic strategies for selecting $V$ have also been proposed. For example, Zhang & Williamson (2020) recommend fitting a mixture model to the data and estimating $V$ from the posterior distribution of the mixture. Alternatively, Bayesian optimization may be employed to identify an optimal choice of $V$ in a data-driven manner.

[5]A high-level implementation will be provided within a well-known causal discovery package.

error. We compare KCI and *FastKCI* in different configurations with a growing violation of $H_0$ (i.e., $\sigma_{\text{vio}}^2$ is increasing). Figure 3 displays that both approaches have similar performance at a sample size of $n = 1200$.

For additional comparison, we consider the setting from Section 5.1, but now $X$ directly causes $Y$. We calibrate the setting to a small violation (approximately $1/3$ of the signal), under which we observe a non-zero type II error. Table 1 shows the power under different numbers of Gaussians $V_{\text{true}}$ in the DGP.

## 5.3 CAUSAL DISCOVERY

We compare the performance of the PC algorithm using *FastKCI* with KCI in causal discovery tasks. For this, we consider two different settings, setting A is derived from Zhang et al. (2012). We sample 6 random variables $\{X_1, \ldots, X_6\}$. For $j > i$ we sample edges with probability 0.3. Based on the resulting DAG, we sample descendants from a Gaussian Process with mean function $\sum_{i \in \text{Pa}(X_j)} \nu_i \cdot X_i$ (with $\nu_i \sim \mathcal{U}[-2, 2]$) and a covariance kernel consisting of a Gaussian kernel plus a noise kernel. Setting B is derived from Liu et al. (2024) and similarly consists of 6 random variables. For $j > i$, we sample edges with probability 0.5. The link function $f_i$ is randomly chosen between being linear and non-linear. For linear components, the edge weights are drawn from $\mathcal{U}[-1.5, -0.5], [0.5, 1.5]$, while non-linear components follow multiple functions (`sin`, `cos`, `tanh`, `sigmoid`,`polynomial`). Added noise is simulated from $\mathcal{N}(0, \sigma_i^2)$ with $\sigma \in \{0.2, 0.5\}$. As shown in Figure 4, both methods exhibit similar performance in precision, recall and F1 score between discovered edges and true causal skeleton.

## 5.4 SCALABILITY

As Figure 5 highlights, *FastKCI* shows excellent scalability, particularly when allowing the number of components $V$ to grow with sample size, which is in accordance with our theoretical consideration. To further investigate the scalability of *FastKCI*, we scale up the sample size in the experiments and report precision, recall, F1 and computation time.[6] The results in Table 2 show that we achieve good results in feasible time even for sample sizes where the traditional KCI fails due to memory and CPU constraints.

## 5.5 COMPARISON TO RANDOMIZED CONDITIONAL INDEPENDENCE TEST

Another recently proposed method for speeding up the KCI is the Randomized Conditional Independence Test (RCIT) (Strobl et al., 2018). The authors show that their approximation of the KCI null by random Fourier features is able to achieve significant speed-ups while being comparable to KCI in a

---

[6]To reduce complexity, in these experiments we approximate the kernel bandwidth instead of determinating it exactly with GP. See Zhang et al. (2012) for detail.

| DGP | n | Method | Precision | Recall | F1 | Time [s] |
|---|---|---|---|---|---|---|
| Gaussian Process | 2000 | KCI | 0.9833 | 1.0000 | 0.9910 | 467.48 |
| | | FastKCI(V=3) | 0.9526 | 1.0000 | 0.9740 | 250.59 |
| | | FastKCI(V=10) | 0.8980 | 0.9500 | 0.9213 | 129.40 |
| | 10000 | KCI | 0.9130 | 1.0000 | 0.9496 | 22240 |
| | | FastKCI(V=3) | 0.9130 | 1.0000 | 0.9496 | 9136.5 |
| | | FastKCI(V=10) | 0.9167 | 1.0000 | 0.9524 | 1505.2 |
| Nonlinear Process | 2000 | KCI | 0.9704 | 0.9299 | 0.9470 | 1098.9 |
| | | FastKCI(V=3) | 0.9783 | 0.9251 | 0.9484 | 587.72 |
| | | FastKCI(V=10) | 0.9641 | 0.8861 | 0.9196 | 219.73 |
| | 10000 | KCI | 1.0000 | 1.0000 | 1.0000 | 99360 |
| | | FastKCI(V=3) | 1.0000 | 0.9711 | 0.9847 | 51709 |
| | | FastKCI(V=10) | 1.0000 | 0.9841 | 0.9916 | 12152 |
| | 20000 | FastKCI(V=10) | 1.0000 | 1.0000 | 1.0000 | 68086 |
| | | FastKCI(V=50) | 0.9500 | 0.9722 | 0.9575 | 1909.7 |
| | 50000 | FastKCI(V=50) | 0.9667 | 0.9818 | 0.9739 | 74095 |
| | | FastKCI(V=100) | 0.9818 | 0.9533 | 0.9664 | 7959.4 |
| | 100000 | FastKCI(V=100) | 1.0000 | 1.0000 | 1.0000 | 86342 |
| | | FastKCI(V=200) | 0.8611 | 1.0000 | 0.9251 | 12253 |

Table 2: Precision, recall, F1 and computational time of the KCI and *FastKCI* on very large samples.

wide range of DGPs. We demonstrate that our method that *exactly* replicates the KCI null law instead of approximating it, has an advantage on both type-I and type-II error when it comes to a complex, multi-modal DGP with $V = 10$ Gaussians in the ground-truth. See result for type-I-error in Table 3 and for type-II in Table 8 in Appendix A.2.

| | FastKCI ($V = 10$) | | | KCI | | | RCIT | | |
|---|---|---|---|---|---|---|---|---|---|
| $|Z|$ | Type-I Error ($\alpha = 1\%$) | Type-I Error ($\alpha = 5\%$) | CPU time [s] | Type-I Error ($\alpha = 1\%$) | Type-I Error ($\alpha = 5\%$) | CPU time [s] | Type-I Error ($\alpha = 1\%$) | Type-I Error ($\alpha = 5\%$) | CPU time [s] |
| 1 | 0.005 | 0.085 | 3.56 | 0.01 | 0.03 | 19.51 | 0.995 | 1 | 0.016 |
| 3 | 0.01 | 0.065 | 6.83 | 0.03 | 0.06 | 118.22 | 0.715 | 0.85 | 0.016 |
| 5 | 0.005 | 0.04 | 10.55 | 0.03 | 0.06 | 177.66 | 0.265 | 0.39 | 0.016 |
| 7 | 0.015 | 0.05 | 22.03 | 0.005 | 0.065 | 163.48 | 0.055 | 0.135 | 0.016 |
| 10 | 0.005 | 0.045 | 43.02 | 0.01 | 0.02 | 320.21 | 0.03 | 0.11 | 0.016 |
| 12 | 0.015 | 0.05 | 63.4 | 0.02 | 0.07 | 323.3 | 0.13 | 0.275 | 0.017 |
| 15 | 0.01 | 0.04 | 79.31 | 0 | 0.03 | 669.18 | 0.18 | 0.37 | 0.017 |
| 30 | 0.005 | 0.035 | 284.63 | 0.005 | 0.05 | 800.66 | 0.35 | 0.54 | 0.018 |

Table 3: Type-I Errors for FastKCI, KCI and RCIT in a process with 10 Gaussians in the ground-truth.

## 5.6 COMPUTATION TIME

We showcased the empirical performance of *FastKCI* for both pure CI tasks as well as causal discovery with PC. To shed light on the computational speed-up, we report the computation time in Figure 5, which – depending on the choice of the main tuning parameters $V$ and $J$ – is significantly faster than for the KCI.

## 6 APPLICATION TO PRODUCTION DATA

We apply our proposed method to a semi-synthetic dataset for causal discovery, contained in `causal-assembly` (Göbler et al., 2024). The ground-truth consists of 98 production stations, each dedicated to specific automated manufacturing processes where individual components are progressively assembled. The processes involved, such as press-in and staking, are mechanically complex and non-linear. The resulting data provides a real-world example on which causal discovery can enhance the understanding of causes for production results and yields a good benchmark for our proposed methodology.

We compare KCI with the *FastKCI* variant, setting $V = 10$. Table 4 depicts the full assembly line with 98 nodes, while Table 5 shows results only for one of the production stations with 16 nodes. Please refer to Göbler et al. (2024) for an overview of results with alternative CD algorithms such as PC with fisher-Z, lingam and others. In terms of precision and recall, KCI performs slightly better than *FastKCI*, but both outperform all tested methods in the original benchmark paper. In terms of computational speed, *FastKCI*, especially in the station setting with a smaller number of nodes, has a significant edge over the standard method.

| Type | Mean Precision | Mean Recall | Mean F1 | Mean Computation Time [s] |
|---|---|---|---|---|
| PC (FastKCI) | 0.6263 | 0.0951 | 0.1651 | 21107 |
| PC (KCI) | 0.5894 | 0.1325 | 0.2163 | 24659 |
| snr | 0.3501 | 0.1311 | 0.0926 | |
| grandag | 0.3193 | 0.0109 | 0.0045 | |
| lingam | 0.3281 | 0.1092 | 0.0721 | |
| PC (Fisher-Z) | 0.4121 | 0.1170 | 0.0968 | |
| notears | 0.5209 | 0.0978 | 0.1019 | |
| das | 0.2971 | 0.0784 | 0.0474 | |

Table 4: Mean result of 10 repetitions on the Causal Assembly benchmark with $n = 1000$.

| Type | Mean Precision | Mean Recall | Mean F1 | Mean Computation Time [s] |
|---|---|---|---|---|
| FastKCI (V=10) | 0.7815 | 0.8167 | 0.7982 | 371.5 |
| KCI | 0.7751 | 0.8847 | 0.8249 | 1230.1 |

Table 5: Mean result of 100 repetitions on the Causal Assembly benchmark, Station 3, with $n = 2000$.

## 7 CONCLUSION AND LIMITATIONS

Our paper introduced *FastKCI*, which turns the cubic-time KCI into a more scalable and embarrassingly parallel procedure. The clustering approach in the conditioning set $Z$ via a mixture of experts remains statistically valid under mild conditions, while the per test cost falls roughly by a factor $1/V^2$. Our experiments show that *FastKCI* preserves KCI's Type-I and Type-II error across diverse settings and cuts wall-clock time for both CI and causal discovery dramatically as sample size grows. Particularly, we showed empirically how FastKCI can scale causal discovery by letting the number of components $V$ grow with sample size, even under violation of the MoE assumption. If $V$ grows such that average cluster size stays constant, FastKCI approaches linearity in $n$ in computation time. Thus, our method is a promising approach for scaling up CI testing.

The main limitation is reliance on the Gaussian-mixture assumption for partitioning, although it is a common assumption for multimodal settings and we showed FastKCIs robustness under its violation, misleading clusters can hurt power, particularly under small $J$. Further, our work does not address the issue of large conditioning set sizes $|Z|$, but it would be worth investigating a possible combination of our framwork with CI tests for large $|Z|$ (e.g., Bellot & van der Schaar (2019)). Future work could also include exploring alternative sample partitioning and importance weighting schemes in order to further refine efficiency of FastKCI.

## 8 REPRODUCIBILITY STATEMENT

We provide full source code and instructions in the supplementary material to reproduce all experiments, including simulations, and causalAssembly benchmarks. The data-generating processes for synthetic experiments are fully specified, and the semi-synthetic benchmark dataset (causal-assembly) is publicly available. Runtime environments, and computing resources are documented in Appendix B. The hyperparameters used in the experiments are specified in the according sections. The implementation of FastKCI is included and will also be released as part of an open-source causal discovery package to facilitate use by the community.

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

# A    APPENDIX

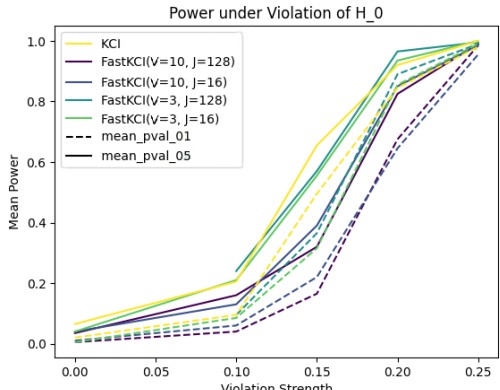

Figure 3: Power of *FastKCI* in different configurations compared to KCI. The violation of the null-hypothesis is increasing on the x-axis.

| $V_{\text{true}}$ | Algorithmn | Power ($\alpha = 5\%$) | Power ($\alpha = 1\%$) |
|---|---|---|---|
| 1 | FastKCI(V=10, J=128) | 0.67 | 0.43 |
| | FastKCI(V=10, J=16) | 0.665 | 0.405 |
| | FastKCI(V=3, J=128) | 0.86 | 0.745 |
| | FastKCI(V=3, J=16) | 0.87 | 0.665 |
| | KCI | 0.91 | 0.77 |
| 3 | FastKCI(V=10, J=128) | 0.725 | 0.545 |
| | FastKCI(V=10, J=16) | 0.79 | 0.62 |
| | FastKCI(V=3, J=128) | 0.905 | 0.74 |
| | FastKCI(V=3, J=16) | 0.89 | 0.735 |
| | KCI | 0.86 | 0.72 |
| 10 | FastKCI(V=10, J=128) | 0.73 | 0.55 |
| | FastKCI(V=10, J=16) | 0.755 | 0.54 |
| | FastKCI(V=3, J=128) | 0.87 | 0.645 |
| | FastKCI(V=3, J=16) | 0.82 | 0.665 |
| | KCI | 0.855 | 0.605 |

Table 1: Power of KCI and *FastKCI* under violation of $H_0$. $X$ and $Y$ are confounded by $Z$, but there is also a direct edge between them.

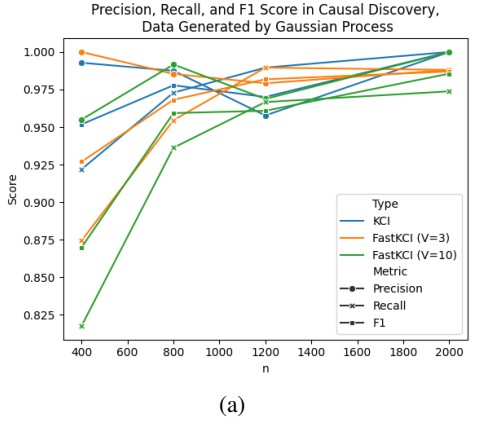

(a)

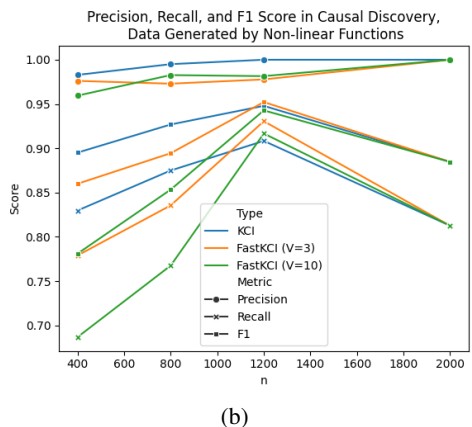

(b)

Figure 4: Precision, recall and F1-Score for KCI and *FastKCI* in causal discovery with growing sample size in setting A (left) and setting B (right).

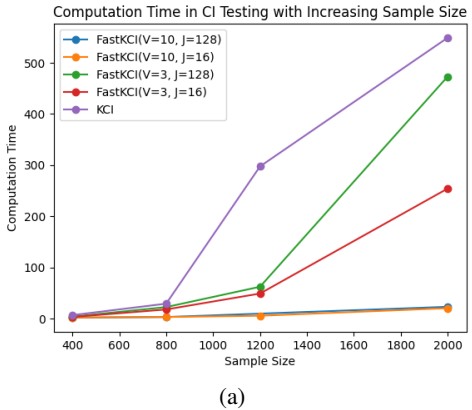

(a)

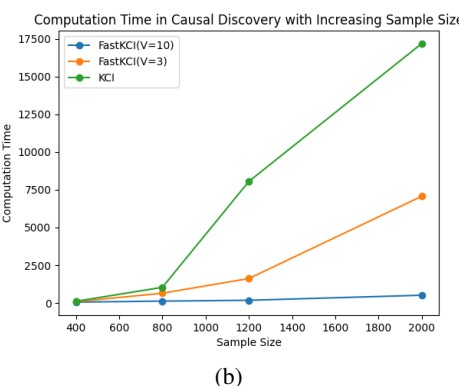

(b)

Figure 5: Computation time with increasing sample size for (a) conditional independence testing and (b) causal discovery with the PC algorithm.

## A.1 ABLATION ON HYPERPARAMETER $V$

We perform an ablation study on the hyperparameter $V$. For this, we use two DGPs, one that follows Assumption 1, precisely the one from Section 5.1, with a mixture of $V = 10$ Gaussians, and one, that violates it[7].

We find that the results are relatively insensitive to the choice of $V$. Small ratios of $V/n$ tend to have high computational complexity. Very large ratios, e.g. the case $V = 100$ and $n = 1200$ appear to not approximate the conditioning set well enough anymore and thus are not recommended. They further can increase computation time again because of inbalanced component sizes and unstable computations on small samples. Anything inbetween is recommendable.

| | $n = 1200$ | | | $n = 5000$ | | |
|---|---|---|---|---|---|---|
| $V$ | Type-I Error ($\alpha = 1\%$) | Type-I Error ($\alpha = 5\%$) | CPU time [s] | Type-I Error ($\alpha = 1\%$) | Type-I Error ($\alpha = 5\%$) | CPU time [s] |
| 3 | 0.005 | 0.050 | 48.73 | 0.005 | 0.025 | 1133.64 |
| 10 | 0.005 | 0.035 | 3.67 | 0.015 | 0.050 | 272.55 |
| 20 | 0.005 | 0.040 | 2.47 | 0.010 | 0.045 | 58.16 |
| 50 | 0.010 | 0.075 | 3.40 | 0.000 | 0.035 | 11.23 |
| 100 | 0.015 | 0.185 | 5.69 | 0.005 | 0.035 | 9.91 |

Table 6: Ablation on the choice of $V$ in FastKCI on the tests performance in a DGP with a mixture of 10 Gaussians.

| | **n = 1200** | | | **n = 5000** | | |
|---|---|---|---|---|---|---|
| $V$ | Type-I Error ($\alpha = 1\%$) | Type-I Error ($\alpha = 5\%$) | CPU time [s] | Type-I Error ($\alpha = 1\%$) | Type-I Error ($\alpha = 5\%$) | CPU time [s] |
| 3 | 0.000 | 0.045 | 45.49 | 0.015 | 0.045 | 1140.41 |
| 10 | 0.010 | 0.075 | 3.43 | 0.020 | 0.080 | 276.25 |
| 20 | 0.000 | 0.040 | 2.22 | 0.015 | 0.085 | 58.89 |
| 50 | 0.010 | 0.095 | 3.04 | 0.035 | 0.090 | 10.85 |
| 100 | 0.020 | 0.155 | 5.01 | 0.025 | 0.100 | 9.55 |
| **KCI** | 0.010 | 0.040 | 32.84 | | | |

Table 7: Ablation on the choice of $V$ in FastKCI on the tests performance in a DGP with a conditioning set consisting of a non-Gaussian distribution.

Particularly, these experiments also demonstrate how FastKCIs performance appears to be not heavily reliant on Assumption 1. Both under misspecification of the number of mixture components $V$ and under a completely non-Gaussian conditioning set we observe that Type-I errors remain intact.

## A.2 POWER COMPARED TO RCIT

We compare the power of FastKCI to KCI and RCIT in a study similar to section 5.2. We see, that the type-II error of FastKCI is much more competitive to KCI than the one of RCIT. This further underlines our argument that in certain processes approximating the null can be insufficient.

| | **FastKCI** | | **KCI** | | **RCIT** | |
|---|---|---|---|---|---|---|
| Violation strength | Type-II Error ($\alpha = 5\%$) | Type-II Error ($\alpha = 1\%$) | Type-II Error ($\alpha = 5\%$) | Type-II Error ($\alpha = 1\%$) | Type-II Error ($\alpha = 5\%$) | Type-II Error ($\alpha = 1\%$) |
| 0.10 | 0.960 | 0.840 | 0.905 | 0.795 | 0.975 | 0.915 |
| 0.15 | 0.835 | 0.680 | 0.505 | 0.340 | 0.940 | 0.810 |
| 0.20 | 0.325 | 0.175 | 0.155 | 0.080 | 0.690 | 0.480 |
| 0.25 | 0.015 | 0.010 | 0.015 | 0.000 | 0.380 | 0.190 |

Table 8: Type-II error comparison of KCI, FastKCI and RCIT for a process with a mixture of 10 Gaussians. The violation strength refers to the signal of the violation relative to the signal of the influence of $Z$ on $X$ and $Y$.

---

[7]We draw $U \sim \text{Unif}(-\pi, \pi)$ and set $Z = \sin(U) + 0.3\sin(3U) + 0.1U^2$. Given $Z$, we generate $X = f_X(Z) + \varepsilon_X$ and $Y = f_Y(Z) + \varepsilon_Y$, with independent noises $\varepsilon_X, \varepsilon_Y \sim \mathcal{N}(0, 0.5I)$, ensuring $H_0$ to hold while introducing post-link-noise. We use non-linear but distinct mappings (e.g., random linear projections followed by $\tanh$ or cubic terms).

## A.3 ADDITIONAL RESULTS

Here we provide additional results for the other stations of the causal assembly benchmark.

| Type | Mean Precision | Mean Recall | Mean F1 | Mean Computation Time [s] |
|---|---|---|---|---|
| FastKCI (V=10) | 1.000 | 0.4314 | 0.5938 | 25.369 |
| KCI | 0.9891 | 0.6814 | 0.7994 | 235.12 |

Table 9: Mean result of 100 repetitions on the causal assembly benchmark, Station 1, with $n = 2000$.

| Type | Mean Precision | Mean Recall | Mean F1 | Mean Computation Time [s] |
|---|---|---|---|---|
| FastKCI (V=10) | 0.6239 | 0.3240 | 0.4262 | 4427.1 |
| KCI | 0.5963 | 0.3980 | 0.4772 | 25279 |

Table 10: Mean result of 100 repetitions on the causal assembly benchmark, Station 2, with $n = 2000$.

| Type | Mean Precision | Mean Recall | Mean F1 | Mean Computation Time [s] |
|---|---|---|---|---|
| FastKCI (V=10) | 0.5236 | 0.3499 | 0.4190 | 3282.8 |
| KCI | 0.5536 | 0.4665 | 0.5057 | 21755 |

Table 11: Mean result of 100 repetitions on the causal assembly benchmark, Station 4, with $n = 2000$.

## A.4 KERNEL-BASED CONDITIONAL INDEPENDENCE TEST (KCI)

Here we provide a clearly structured algorithm box summarizing the original Kernel-based Conditional Independence (KCI) test introduced by Zhang et al. (2012) for clarity and comparison with the fast procedure.

# B IMPLEMENTATION DETAILS

## B.1 PARITIONING SCHEME

In our implementation, we do not run a full EM algorithm for the Gaussian mixture. Instead, for each of the $j \in J$ partitioning rounds we use a simple empirical-Bayes sampling scheme over $Z$ approximating the NIW prior: we draw $V$ component means from a Normal centered at the empirical mean of $Z$, use an identity covariance for all components, draw mixture weights from a symmetric Dirichlet prior $\mathrm{Dir}(\alpha)$, with $\alpha = 500$ and then sample cluster assignments for each $z_i$ from the resulting categorical distribution (proportional to $\pi_v \mathcal{N}(z_i \mid \mu_v, I)$). The goal of this step is to generate reasonable local partitions of the conditioning set $Z$, the importance-weighted aggregation then corrects for randomness across partitions. This lightweight scheme is sufficient in practice and avoids the extra cost of running EM inside each repetition.

## B.2 AGGREGATION SCHEME

To compute the weights $\ell^{(j,v)}$, we place independent Gaussian process regression models on the local relations $X \mid Z$ and $Y \mid Z$ within each cluster $v$. Concretely, for $i \in \mathcal{C}_v^{(j)}$, we assume $X_i = f_{X,v}(Z_i) + \varepsilon_{X,i}$ and $Y_i = f_{Y,v}(Z_i) + \varepsilon_{Y,i}$, with $f_{X,v}, f_{Y,v} \sim \mathrm{GP}(0, k_\theta)$ and Gaussian noise $\varepsilon_{\cdot,i} \sim \mathcal{N}(0, \sigma^2)$. Conditional on $Z_{\mathcal{C}_v^{(j)}}$ and partition indicator $U^{(j)}$, the block of observations is therefore jointly Gaussian, e.g.

$$P(X_{\mathcal{C}_v^{(j)}} \mid Z_{\mathcal{C}_v^{(j)}}, U^{(j)}) = \mathcal{N}(0, K_X^{(j,v)} + \sigma_X^2 I),$$

---

[8]For simplicity, we refer for details to Zhang et al. (2012). Sampling from the null distribution involves multiple eigenvalue and vector computations.

| Type | Mean Precision | Mean Recall | Mean F1 | Mean Computation Time [s] |
|---|---|---|---|---|
| FastKCI (V=10) | 0.4619 | 0.5350 | 0.4953 | 543.38 |
| KCI | 0.4822 | 0.6130 | 0.5390 | 2550.6 |

Table 12: Mean result of 100 repetitions on the causal assembly benchmark, Station 5, with $n = 2000$.

---

**Algorithm 2** Kernel-based Conditional Independence Test (KCI) by Zhang et al. (2012)

---

**Require:** Datasets $\{x_i, y_i, z_i\}_{i=1}^n$ where $x_i \in \mathbb{R}^{d_x}$, $y_i \in \mathbb{R}^{d_y}$, $z_i \in \mathbb{R}^{d_z}$; kernel functions $k_x$, $k_y$, and $k_z$ for $X, Y$, and $Z$ respectively; regularization parameter $\lambda$.

**Ensure:** Test decision for $H_0 : X \perp Y \mid Z$

1: Aggregate $\ddot{X} = (X, Z)$. Compute the kernel matrices $K_{\ddot{X}}$, $K_Y$, and $K_Z$ for the datasets using the respective kernel functions:

$$(K_{\ddot{X}})_{ij} = k_x(\ddot{x}_i, \ddot{x}_j), \quad (K_Y)_{ij} = k_y(y_i, y_j), \quad (K_Z)_{ij} = k_z(z_i, z_j)$$

2: Center the kernel matrices $K_{\ddot{X}}$, $K_Y$ and $K_Z$:

$$\tilde{K}_{\ddot{X}} = H K_{\ddot{X}} H, \quad \tilde{K}_Y = H K_Y H \quad \tilde{K}_Z = H K_Z H$$

where $H = I - \frac{1}{n} \mathbf{1} \mathbf{1}^T$ and $\mathbf{1}$ is a column vector of ones.

3: Calculate the projection matrix from a kernel ridge regression on $Z$:

$$R_Z = I - \tilde{K}_Z (\tilde{K}_Z + \lambda I)^{-1} = \lambda(\tilde{K}_Z + \lambda I)^{-1}$$

4: Compute the residual kernels of $\ddot{X}$ and $Y$ after conditioning on $Z$:

$$\tilde{K}_{\ddot{X}|Z} = R_Z \tilde{K}_X R_Z, \quad \tilde{K}_{Y|Z} = R_Z \tilde{K}_Y R_Z$$

5: Compute the test statistic:

$$T_{CI} \triangleq \frac{1}{n} \text{Tr}\left(\tilde{K}_{\ddot{X}|Z} \tilde{K}_{Y|Z}\right)$$

6: Bootstrap samples from the asymptotic distribution of the test statistic $\check{t}_{CI}$[8]

7: Compute the p-value:

$$p = \frac{1}{B} \sum_{b=1}^{B} \mathbb{I}(\check{t}_{CI}^b \geq T_{CI})$$

8: **return** p-value for $H_0$

---

and analogously for $Y$.

In the implementation, $\ell^{(j,v)}$ is computed exactly as the sum of the log-marginal likelihoods for $X \mid Z$ and $Y \mid Z$ on each block, and the weights $w_j$ are obtained via a softmax over the resulting partition log-likelihoods.

## C  COMPUTING RESOURCES.

The experiments in Sections 5 and 6 were performed on a high-performance computing cluster. Each node has two Intel Xeon E5-2630v3 with $8$ cores and a $2, 4$GHz frequency as well as $64$GB RAM. For the results with a higher number of $J$, multiple nodes where used. While the runtime of single repetitions of CI or PC can be derived from Section 5.6 or respective columns in the result tables (e.g., column "Time" in Table 2), the full runtime of reproducing all experiments can be estimated around two to four weeks on a single node.

## D  STATEMENT ON AI USAGE

For this research paper, large language models were used solely to assist with literature search, writing, and coding. All conceptualization, ideation, and theoretical contributions were carried out without AI support. The paper and code were authored entirely by the researchers, with AI serving only as a support and feedback tool.

