# OpenReview forum: "A Fast Kernel-based Conditional Independence Test with Application to Causal Discovery"
_ICLR.cc/2026/Conference — Submitted to ICLR 2026_

### Official Review · Reviewer_5CwZ · 2025-10-27

**Soundness:** 3
**Presentation:** 3
**Contribution:** 3
**Rating:** 4
**Confidence:** 4

**Summary:**

This paper addresses the cubic-time computational cost of the Kernel-based Conditional Independence Test (KCIT). The proposed method, FastKCI, introduces a scalable and embarrassingly parallel framework that partitions the data based on a Gaussian mixture model over the conditioning variable $Z$. For each of $J$ independently sampled partitions, the method performs local KCI tests within $V$ clusters and aggregated the resulting statistics via importance weighting. Under mild assumptions, the approach preserves the null distribution of the original KCIT while reducing the effective runtime to approximately $O(Jn^3/V^2)$. Empirical results on synthetic and causal discovery benchmarks show that FastKCI achieves comparable test accuracy while substantially improving computational efficiency.

**Strengths:**

1.	Significance: The paper targets the computational bottleneck of KCIT, which is a crucial component in constraint-based causal discovery. Scaling KCI to large dataset is an important and timely problem, and the paper’s framework offers a novel and clear way to parallelize and accelerate the test.
2.	The method is well-aligned with existing theory and preserves statistical validity. FastKCI maintains the null distribution of the original KCIT under mild conditions, and the derivation is clearly explained.
3.	Quality: The experiments are relatively thorough. The empirical evaluation demonstrates that the method achieves similar Type-I and Type-II error performance to KCI, while reducing the runtime. The scenarios considered are generally appropriate, and the results are consistent with the claims.

**Weaknesses:**

1.	Some modeling and notation choices are not explained in sufficient detail. For example, in Assumption 1, the Normal-Inverse-Wishart prior is introduced, but there is no introduction to the hyperparameters and no guidance on how its hyperparameters are selected in practice or how sensitive the method is to these choices. In addition, some presentation issues (e.g., the placement of Table 2 below the footnote) hinder readability.
2.	The paper provides a discussion on how to choose the variable $J$ and also gives some recommendations for choosing $V$. However, it might be practically more useful to choose $V$ according to the sample size, e.g., making $V$ a function of the sample size $n$. It would be helpful if the authors could clarify whether this is feasible.
3.	Not enough assessment with respect to the performance comparison between other baselines and FastKCIT (see question part below).

**Questions:**

1.	In the related work section, several approaches for accelerating KCIT or designing fast conditional independence tests are mentioned. However, the experimental evaluation only includes KCIT and RCIT in a subset of settings. Given that the main claim of the paper is scalability while preserving statistical performance, a broader comparison against these established fast CI tests would be necessary to support the empirical contribution. I suggest including more methods to more fully substantiate the claimed advantages.
2.	More simulation results appear to be based on data generated from Gaussian mixture distributions of $Z$, which align with the modeling assumption. To better understand robustness, could the authors provide results on data where $Z$ is not well modeled by Gaussian mixtures (beyond the production dataset), or discuss expected behavior in such cases?
3.	For the comparison with RCIT, could the authors provide the detailed data-generating process used in Section 5? Since RCIT often performs well in practice, a clearer description of the experimental setup would clarify under which conditions FastKCI has an advantage.
4.	In Table 4, FastKCI shows strong precision performance (with a loss in recall). Could the authors provide intuition or explanation for this behavior, especially given that the performance differences in earlier synthetic experiments were relatively small?

---

> ### Author Response · Authors · 2025-11-26
>
> Thank you very much for your time and feedback. We read it carefully and would like to address your questions and concerns.
>
> **W1:** Thanks for pointing out these minor presentational issues. We gladly fix them in a revised version.
>
> **W2:** We agree that from a more practical perspective and in the spirit of approximating a Gaussian Mixture Model instead of finding the “truth” for $V$ it might be practical to make $V$ a function of $n$. Growing $V$ with sample size would also approach linear computation time. We clarify this further in the ablation on $V$.
>
> **Q1:** Thank you very much for this suggestion. The main reason for the lack of comparisons is the absence of implementations. We highlight that our method will be available in a very easy-to-use implementation, which is unfortunately missing for many methods previously proposed. It has been a standard in these papers to compare against KCIT as a benchmark. However, it would be very interesting future work, to have a large-scale simulation study of many KCI speedups across a variety of settings.
>
> **Q2:** Thanks for making this very valuable suggestion. In Appendix A.1 we provide further simulation results of a process that is non-Gaussian (see footnote 7).
>
> **Q3:** Thank you for this suggestion. We will add a more detailed description to Section A.2. Namely:
>
> * First, we draw $Z \in \mathbb{R}^{d_Z}$ with $d_Z = 30$ from a mixture of $n_{\text{components}} = 10$ Gaussians.   Concretely, we sample mixture weights $\pi \sim \text{Dirichlet}(1)$, assign $n\pi_k$ points to component $k$, and for each component draw $Z_i \sim \mathcal{N}(\mu_k, \Sigma_k)$, where $\mu_k$ has i.i.d. $\mathcal{N}(0,3^2)$ entries and $\Sigma_k$ is diagonal with entries uniform in $[0.5,2.0]$. We then shuffle the rows of $Z$.
> * Next, we define nonlinear functions of $Z$: $ f(Z) = \tanh(Z w_{f1}) w_{f2},  g(Z) = \sin(Z w_{g1}) w_{g2}$,  with $w_{f1}$, $w_{f2}$, $w_{g1}$, and $w_{g2}$ i.i.d. standard normal weights.
> * We then generate $X = f(Z) + \varepsilon_X, Y = g(Z) + \varepsilon_Y$,  with $\varepsilon_X, \varepsilon_Y \sim \mathcal{N}(0, 0.1^2)$ i.i.d..
>
> In this setup, (X) and (Y) depend on (Z) through different highly nonlinear, multimodal mappings in a 30-dimensional space, which is a challenging regime for regression-based CI tests such as RCIT. Kernel-based methods like KCI/FastKCI can better capture such complex structure, so this DGP highlights a scenario where FastKCI has an advantage.
>
> **Q4:** In Table 4 we evaluate PC on the full **98**-node assembly line, which is a much harder setting than the small synthetic graphs in Sec. 5.3. There, PC(FastKCI) shows slightly higher precision but lower recall than PC(KCI).
> Intuitively, FastKCI is a bit more **conservative** at the chosen significance level:
>
> * The partitioning in $Z$ reduces the effective sample size within each local KCI, so very weak dependencies are more likely to be treated as (conditionally) independent.
> * In the PC algorithm this means **more edges are pruned** (or not added) when the evidence is borderline.
>
> As a result, PC(FastKCI) might have fewer false positives, leading to a higher precision but also more missed true edges which explains the lower recall, while still being competitive in F1 and clearly faster.
>
> In the earlier synthetic experiments the graphs are small and conditioning sets moderate, so this mild conservativeness has little impact and FastKCI/KCI perform almost identically.
>
> Thanks again for your time!

---

### Official Review · Reviewer_KdRF · 2025-10-28

**Soundness:** 3
**Presentation:** 2
**Contribution:** 2
**Rating:** 4
**Confidence:** 4

**Summary:**

This paper proposes FastKCI, a scalable and parallelizable variant of the Kernel-based Conditional Independence (KCI) test designed to overcome the computational bottleneck of KCI without sacrificing its statistical power. FastKCI's approach is based on a "mixture-of-experts" model and parallel inference techniques. The dataset is partitioned into $V$ smaller subsets, and the partitioning is guided by fitting a Gaussian mixture model (GMM) to the conditioning variables $Z$. A standard KCI test is then conducted on each of these smaller partitions independently and in parallel. Finally, the results from all local tests are combined using an "importance-weighted sampling scheme" to produce the final global test statistic and p-value. The method is validated on synthetic datasets and applied to real-world data for causal discovery. The results confirm that FastKCI achieves comparable accuracy to KCI in causal discovery tasks but in a fraction of the time.

**Strengths:**

1. The paper has a clear motivation and is structured.
2. CI tests lie in the computational bottleneck in constraint-based causal discovery.

**Weaknesses:**

1. The reliability of FastKCI heavily depends on the Gaussian Mixture Model (GMM) assumption. If the true distribution of $Z$ is not well approximated by a GMM, the performance of FastKCI may significantly degrade.
2. The current simulation experiments are conducted only on data generated from Gaussian mixture models. Additional experiments on data that violate the GMM assumption are necessary to assess the robustness of FastKCI under model misspecification.
3. As a practical improvement, it would be useful to include a preliminary goodness-of-fit test. For instance, if the fitted GMM poorly represents
$Z$, the algorithm could either alert the user or automatically adjust the number of partition samples $J$ to mitigate the mismatch.
4. FastKCI introduces two hyperparameters—$V$ (the number of mixture components) and $J$ (the number of partition sampling rounds)—yet the paper provides insufficient guidance on how to select these parameters in practice.
5. From a methodological perspective, the paper’s novelty appears somewhat limited, as the main contribution lies in transferring an existing idea (embarrassingly parallel inference) to the conditional independence testing setting. Nevertheless, the adaptation is technically sound and potentially useful in practice.
6. Despite the claimed efficiency, FastKCI still exhibits cubic computational complexity with respect to the sample size $n$.

**Questions:**

1. Would it be possible to design an optimization criterion or data-driven procedure to automatically determine the optimal values of $V$ and $J$?
2. What are the key advantages of adopting embarrassingly parallel inference techniques compared to kernel acceleration approaches such as incomplete Cholesky decomposition, random Fourier features, or the Nyström approximation?
3. Could the authors clarify how the p-value is computed in FastKCI? For example, is it based on an asymptotic null distribution, permutation testing, or another resampling approach?

---

> ### Author Response · Authors · 2025-11-26
>
> Thank you very much for your time and feedback. We read it carefully and would like to address your questions and concerns.
>
> Concerning **W1** and **W2**, we would like to point out, that even though the reliance on the GMM assumption can be seen as a weakness, we also discuss in the paper that a GMM is a good approximator for a wide range of distributions and we explicitly perform experiments in the main text and the appendix where the data is **not** generated from a GMM and show that FastKCI is robust in these settings.
>
> Regarding your questions:
>
> **Q1:** We provide guidance on Hyperparameter Tuning of $J$ and $V$ in Section 4.3. As explained, $J$ should be chosen as large as available CPU nodes allow. Because $V$ is more challenging, we perform an ablation on it in Appendix A.1. We think it would be challenging, to have data-driven rules determining them, however, we agree that to a certain degree the tuning of $V$ might impact the performance of our method.
>
> **Q2:** Our approach keeps the *exact* KCI test on each block and only changes how computations are organized, not the kernel itself. In contrast, accelerations such as incomplete Cholesky, random Fourier features, or Nyström explicitly approximate the Gram matrix, which can degrade power and require extra hyperparameters (rank, number of features, etc.). By using embarrassingly parallel inference, FastKCI preserves the original RKHS geometry and null distribution of KCI. The importance of this is shown in the comparisons to RCIT in our paper.
>
> **Q3:** FastKCI uses the same spectral null approximation as KCI (Proposition 5 in Zhang et al., 2012), not permutation testing. For each partition we compute the eigenvalues of the relevant covariance operator and simulate many null test statistics as weighted sums of independent $\chi^2_1$ variables; these are then combined across partitions using the importance weights to obtain a global sample from the null distribution. The p-value is the empirical tail probability of the (weighted) FastKCI test statistic under this simulated null.
>
> Thanks again for your review!

---

### Official Review · Reviewer_jwPz · 2025-10-30

**Soundness:** 2
**Presentation:** 2
**Contribution:** 2
**Rating:** 4
**Confidence:** 4

**Summary:**

This paper proposes an accelerated algorithm for the kernel-based conditional independence test (KCI), termed FastKCI. The FastKCI method assumes that the conditioning variable $Z$ follows a Gaussian Mixture Model (GMM), and partitions the original dataset of $n$ samples into $V$ groups according to the $V$ components of the GMM. This reduces the computational complexity from $O(n^3)$ to $O(n^3 / V^2)$. Experimental results demonstrate that FastKCI significantly improves computational efficiency.

**Strengths:**

FastKCI substantially reduces computational time in most cases without significantly compromising the accuracy of the CITs.

**Weaknesses:**

1. The paper merely combines embarrassingly parallel inference with KCI in a straightforward manner, leading to limited novelty.
2. Although FastKCI achieves faster computation, the experimental results show that the accuracy of the CITs decreases in certain cases. Moreover, the paper only provides a coarse theoretical analysis for the asymptotic behavior of FastKCI as $n \to \infty$, which fails to explain the observed loss in accuracy.
3. The experiments compare FastKCI only with other KCI-based baselines, and the experimental settings focus mainly on low-dimensional conditioning variables $Z$ (e.g., Sections 5.1, 5.3). In contrast, recent CIT methods can efficiently handle high-dimensional $Z$ [GCIT, DGCIT, NNLSCIT, …]. This raises an important question: can the embarrassingly parallel inference framework be extended to aggregate the test statistics of other CIT methods?
4. Several parts of the paper are not clearly articulated; see the Questions section for details.

**Questions:**

1. How are the means, variances, and weights of the different components in the Gaussian Mixture Model estimated? Are they obtained using the EM algorithm? This should be described more explicitly in the paper.
2. I suggest that the authors carefully check the relationship between $X$ and $Y$ across the $V$ different subsets. That’s because the CI relationships in different clusters may change, especially in real data. If Z indeed follows the GMM assumption, do $X$ and $Y$ maintain a consistent conditional independence (CI) relationship across components? Conversely, if $Z$ does not follow the GMM assumption—meaning that the partitioning of samples into $V$ subsets may be not well—how would the CI relationship between $X$ and $Y$ change?
3. In Line 230, what is the exact form of the distribution $P(X^{(j)}, Y^{(j)} | U^{(j)})$ that $l^{(j,v)}$ follows? This requires a more detailed explanation.
4. The experiments shown in Appendix A.1 indicate that the accuracy of FastKCI seems to depend heavily on the correct estimation of the number of components $V$. If we ignore computational speed and focus solely on the accuracy of the CIT, how can one determine an appropriate value of $V$?
5. I would like to know how \textit{Precision} and \textit{Recall} are defined in Tables 2 and 9–11. Specifically, how is the confusion matrix constructed, and how do Precision and Recall relate to Type I and Type II errors?

---

> ### Author Response · Authors · 2025-11-27
>
> Dear Reviewer, thanks a lot for your time and your review. We read it carefully and would like to respond as follows:
>
> **Q1:** Thanks for raising this point. In our implementation we do *not* run a full EM algorithm for the Gaussian mixture. Instead, for each of the $j \in J$ partitioning rounds we use a simple empirical-Bayes sampling scheme over $Z$: we draw $V$ component means from a Normal centered at the empirical mean of $Z$, use an identity covariance for all components, draw mixture weights from a symmetric Dirichlet prior $\text{Dir}(\alpha)$, with $\alpha=500$ and then sample cluster assignments for each $z_i$ from the resulting categorical distribution (proportional to $\pi_v \mathcal{N}(z_i \mid \mu_v, I)$). The goal of this step is to generate reasonable local partitions of the conditioning set $Z$, the importance-weighted aggregation then corrects for randomness across partitions. Since our method only requires plausible partitions rather than maximum-likelihood GMM parameters, this lightweight scheme is sufficient in practice and avoids the extra cost of running EM inside each repetition. We will clarify this as an implementation detail and explicitly state that we do not employ EM in the revised version.
>
> **Q2:** We agree that CI relationships can vary across regions of the conditioning space, especially in real data. Importantly, in FastKCI the Gaussian mixture is only a model for the marginal distribution of $Z$; the latent component index $U \in 1,\dots V$ depends solely on $Z$ (Assumption 1), and within each component we still run the original KCI residualization with the full continuous $Z$ (see Algorithm 1). Thus, if the global null $H_0: X \perp Y \mid Z$ holds, it automatically holds in every mixture component as well, because conditioning on $(Z,U=v)$ for any $v$ cannot re-introduce dependence when $U$ is a function of $Z$. Under alternatives, FastKCI explicitly allows heterogeneity: the local block statistics capture possibly different strengths of $X$–$Y$ dependence in different regions of $Z$, and we aggregate these contributions with importance weights $w_j \propto P(X^{(j)}, Y^{(j)} \mid U^{(j)})$. When the GMM is misspecified, the partitioning may be sub-optimal and can reduce power, but it does not change the target CI notion (we always test $X \perp Y \mid Z$); this is corroborated by our experiments in Appendix A.1, where type-I error and power remain stable even when the true distribution of $Z$ is non-Gaussian.
>
> **Q3:** We agree that this point needs to be stated more explicitly. For a fixed partition $U^{(j)}$ with clusters $C_v^{(j)}$, we place independent Gaussian process regression models on the local relations $X \mid Z$ and $Y \mid Z$ within each cluster $v$. Concretely, for $i \in C_v^{(j)}$, we assume
> $X_i = f_{X,v}(Z_i) + \varepsilon_{X,i}$ and $Y_i = f_{Y,v}(Z_i) + \varepsilon_{Y,i}$, with $f_{X,v}, f_{Y,v} \sim \mathrm{GP}(0,k_{\theta})$ and Gaussian noise $\varepsilon_{\cdot,i} \sim \mathcal{N}(0,\sigma_{\cdot}^2)$.
> Conditional on $Z_{C_v^{(j)}}$ and $U^{(j)}$, the block of observations is therefore jointly Gaussian, e.g.
>
> $P(X_{C_v^{(j)}} \mid Z_{C_v^{(j)}}, U^{(j)}) = \mathcal{N}(0, K^{(j,v)}_{X} + \sigma_X^2 I)$,
>
> and analogously for $Y$.
> We then define
>
> $\ell^{(j,v)} = \log P(X_{C_v^{(j)}} \mid Z_{C_v^{(j)}}, U^{(j)}) + \log P(Y_{C_v^{(j)}} \mid Z_{C_v^{(j)}}, U^{(j)})$,
>
> so that the partition-wise likelihood factorizes as
>
> $\log P(X^{(j)}, Y^{(j)} \mid U^{(j)}) = \sum_v \ell^{(j,v)}$.
>
> In the implementation, $\ell^{(j,v)}$ is computed exactly as the sum of the log-marginal likelihoods for $X \mid Z$ and $Y \mid Z$ on each block, and the weights $w_j$ are obtained via a softmax over the resulting partition log-likelihoods. We will add this explicit GP likelihood form as an implementation detail to clarify $\ell^{(j,v)}$ in the revised manuscript.
>
>
> **Q4:** Appendix A.1 indeed shows that extreme mis-specification of $V$ can affect power. If one is *only* interested in accuracy and not in speed, a conservative choice is to use relatively small $V$ and not let it grow with sample size. In practice, more principled choices of $V$ can be obtained by standard mixture-model selection (e.g. BIC or held-out log-likelihood for the GMM over $Z$), or by the Bayesian and data-driven strategies (see Section 4.3 and Footnote 4).
>
> **Q5:** For Tables 2 and 9–11 we compare the learned graph to the ground-truth graph at the level of a causal skeleton, as typical in recent causal discovery benchmarks. Precision, Recall and F1 are then defined analogously to a binary classification problem.
> In terms of CI testing errors, a high FP rate (many false rejections of $H_0$) tends to reduce Precision, while a high FN rate (many missed dependencies) tends to reduce Recall.

---

### Official Review · Reviewer_9bnz · 2025-10-31

**Soundness:** 3
**Presentation:** 3
**Contribution:** 3
**Rating:** 6
**Confidence:** 3

**Summary:**

The paper is related to causal discovery and proposes a kernel-based Conditional Independence Test (FastKCI), which uses a divide-and-aggregate strategy to reduce the typical cubic numerical complexity in the number of samples. The approach breaks the full kernel into V smaller parts (data partitions). Several numerical experiments are performed to illustrate the performance of the ensemble approach. Results for different baseline setups look promising. Type I error and statistical power remain competitive while saving computational time.

**Strengths:**

S1 The considered problem is interesting and relevant.
S2 The paper is technically sound and overall well-written.
S3 To reduce complexity by 1/V^2 is impressive.
S3 The evaluation and comparison against baselines is convincing.

**Weaknesses:**

W1 The scalability regarding the number of involved variables remains somewhat unclear.
W2 Limitations of the paper’s methods could be better discussed.

**Questions:**

Does it make a difference how the data is partitioned into the V subsets?

Regarding Table 3: Does the approach scale in |Z|? Is the approach still fast if the number of involved variables is 100 or 1000?

How should the hyper parameter V be automatically chosen in general applications? Is the trade-off between accuracy and speed plannable?

The figures and tables are too small. I understand that there is a lack of space but font sizes in figures and tables should be as in the text.

Under which conditions the approach performs not good (e.g. regarding the data partitioning when highly correlated clusters are chosen/realized)?

---

> ### Author Response · Authors · 2025-11-26
>
> Thank you very much for your time and feedback. We read it carefully and would like to address your questions and concerns.
>
> **W2:** Would you have suggestions on how to improve the current discussion of the limitations?
>
> **Q1:** Yes, the way data are partitioned matters. We deliberately partition in the $Z$-space via the Gaussian mixture so that points with similar $Z$ fall into the same component, which induces an approximately block-diagonal kernel structure and preserves the spirit of conditioning on $Z$. Using arbitrary or purely random partitions would still yield valid tests but lose performance via the dependencies that are lost in-between the blocks.
>
> **Q2:** FastKCI mainly improves the dependence on the sample size $n$; like KCI, it still inherits the usual challenges when $|Z|$ becomes very large. Computationally, $|Z|$ enters through the mixture estimation and kernel evaluations, so runtime grows with $|Z|$ but not exponentially. For $|Z|$ in the order of tens (as in our experiments), the method remains fast; for $|Z|$ in the hundreds or even thousands both KCI and FastKCI would typically require additional dimensionality reduction or structure (e.g., sparse kernels) which are beyond our work.
>
> **Q3:** As we discuss in Section 4.3, in practice, $V$ controls the average cluster size $n/V$: small $V$ gives statistics closest to full KCI but less speed-up; larger $V$ yields more speed-up but potentially a mild loss of power if clusters become too small and too much cross-cluster covariance is neglected. Additional experimental insights are provided in the ablation in Appendix A.1 (page 15).
>
> **Q4:** We appreciate this comment. We will increase the font sizes in figures and tables to be closer to the main text in the camera-ready version, as the page limit increases.
>
> **Q5:** FastKCI is less effective when the partitioning does not reflect meaningful structure in $Z$. For example, if the learned mixture components are poorly separated, the kernel matrices are not strongly block-diagonal and the computational gains diminish. Performance can also degrade if clusters become too small relative to $n$, making local KCI estimates noisy, or if the dependence between $X$ and $Y$ given $Z$ is highly nonlocal in $Z$ so that splitting across components breaks informative patterns. In practice, as the `causalAssembly` benchmark and the simulations have shown, FastKCI appears to be relatively insensitive, even if the conditioning set is only approximated by a Gaussian Mixture.
>
> Thanks again for the review and we appreciate your time reading our answers.

---

### Official Review · Reviewer_yocb · 2025-11-11

**Soundness:** 3
**Presentation:** 3
**Contribution:** 3
**Rating:** 6
**Confidence:** 3

**Summary:**

This paper proposes a scalable, kernel-based conditional independence testing approach that first partitions the data into subsets using a Gaussian mixture model, then performs parallel conditional independence tests, and finally aggregates the results using an importance-weighted sampling scheme.

**Strengths:**

-	The paper presents a strong motivation and a practical approach for improving the efficiency of conditional independence tests, and consequently, constraint-based causal discovery.
-	The paper is well-written and clearly presented.

**Weaknesses:**

-	Missing relative references. Please see the question 1 and 2 in Questions section.
-	This paper provides limited evaluation of generality: missing real-world applications, limited settings on synthetic data. Please see questions 5, 6, and 7 in Questions section.

**Questions:**

1.	I believe [1] also partitions the data into subsets, applies parallel kernel tests, and aggregates the results for efficient conditional independence testing. This work is highly relevant and should be discussed in the paper.
> [1] Guan, Zhengkang, and Kun Kuang. "Efficient Ensemble Conditional Independence Test Framework for Causal Discovery." arXiv preprint arXiv:2509.21021 (2025).
2.	How does the computational complexity of fastKCI compare to [1]? [1] claims linear complexity in sample size \(n\). For fastKCI to achieve similar complexity, it requires \(V = n\) components, which may be impractically large.
3.	What is the purpose of assuming a Normal-Inverse-Wishart prior on \(Z\)?
4.	Does the accuracy of the aggregated measurements depend on the reliability of individual measurements? Additionally, will errors in learning the mixture Gaussian distribution affect the accuracy of subsequent steps?
5.	Why do the authors assume nonlinear causal functions? It would also be informative to report results for linear causal functions.
6.	Sample size may not be the dominant factor affecting causal discovery efficiency. The main challenge for constraint-based approaches lies in the accuracy and scalability of conditional independence tests when the conditioning set is high-dimensional. How does the proposed approach perform under such scenarios?
7.	Although the empirical results on semi-synthetic data are promising, it would be valuable to evaluate performance on real-world datasets, such as Sachs et al. (2005).

---

> ### Author Response · Authors · 2025-11-26
>
> Thank you very much for your time and feedback. We read it carefully and would like to address your questions and concerns.
>
> **Q1:** We thank the reviewer for pointing us to [1], which indeed also follows a divide–aggregate strategy in CI. We note that [1] was released on the day of the ICLR deadline and thus was not available to us prior submission. We will add [1] to the related-work section and discuss the connection explicitly.
> Methodologically, however, the two approaches are quite different. E-CIT [1] is a *generic wrapper* that **randomly** partitions the $n$ observations into $K$ subsets of fixed size $n_k$, applies any base CI test (e.g. KCIT, RCIT, Fisher-Z) to each subset, and then combines only the resulting p-values via a stable-distribution based combination rule.
> In contrast, FastKCI is a *KCI-specific* procedure that models the distribution of the conditioning variables $Z$ via a Gaussian mixture and partitions the data in the $Z$-space. This induces an approximately block-diagonal structure in the kernel matrices, and we compute local KCI statistics *and their null distributions* within each mixture component and then aggregate them via importance weights derived from a mixture-of-experts likelihood. As a consequence, FastKCI preserves the original KCI test statistic and its $\chi^2$-mixture null law.
>
> **Q2:** The difference in computation time is that FastKCI still uses the full sample size $n$ for each sampling round $j\in J$, whereas E-CIT purely relies on subsamples of fixed size. As discussed in our paper, if we allow $V$ to grow with sample size, we could also enforce an approximately stable component size $n_V$, approaching linear computation time. E-CIT follows a similar argument, of testing on a fixed subsample size, effectively linearizing the time. Therefore, we follow a more distributional-driven argument in the partition, while E-CIT uses a random divide-and-conquer scheme.
>
> **Q3:** The Normal–Inverse–Wishart prior is used only as a weak, conjugate regularizer for the Gaussian mixture over $Z$. It stabilizes the estimation of cluster means and covariances (especially for small clusters or higher-dimensional $Z$) and gives a well-defined sampling distribution for partitions.
>
> **Q4:** As with any divide–aggregate scheme, the global statistic benefits from reliable local estimates. We partition the data and aggregate the local tests in order to favor accurate partitions following assumption 1. As visible in the ablations on $V$ in the Appendix, FastKCI provides good approximate performance under misspecification of $V$ and Assumption 1.
>
> **Q5:** We focus on nonlinear mechanisms because kernel-based CI tests are particularly advantageous beyond the linear relationships which can be tested with less expensive tests such as the Fisher-Z. The post non-linear model is a typical benchmark for nonparametric causal discovery methods.
>
> **Q6:** We agree that the challenge for constraint-based discovery is twofold: efficient CI testing for large dataset sizes $n$, and for large conditioning set sizes $|Z|$. FastKCI addresses the computational bottleneck of KCI under large $n$ so that it remains usable when many CI tests must be run, while preserving its statistical behavior. Under growing conditioning set sizes $|Z|$, FastKCI scales similar than KCI.
>
> **Q7:** We agree that Sachs et al. (2005) is a very important and widely used benchmark. We focused on `causalAssembly` as empirical real-world data as it is a more recently proposed and very extensive benchmark. Due to the typical sample size of n=853 with 11 nodes and 17 edges, Sachs et al. (2005) is a smaller benchmark than `causalAssembly`.
>
> Thanks again for the review, and we appreciate your time reading our answers.

---

### Author Response · Authors · 2025-12-03

We would like to thank the reviewers and the area chair once more for the careful reading of our work and for the constructive feedback, which has substantially improved the paper. Below we summarize how the main points raised in the review process have been addressed in the revised manuscript:

* **Discussion of Concurrent Work:** We added an explicit discussion of E-CIT to the Related Work section (Section 2). We clarify that while E-CIT shares a divide-and-aggregate philosophy, it is an independent concurrent work that differs substantially from FastKCI in both its random partitioning strategy and its stability-based aggregation scheme.
* **Implementation Details:** We added a new Appendix B to explicitly describe the implementation details requested by reviewers, specifically our empirical-Bayes partitioning scheme and the Gaussian Process-based aggregation mechanism. The implementation will be also publically available and easy-to-use.
* **Computational Complexity:** We refined Section 4.2 to make the complexity statement precise. We explicitly state the per-partition cost of $O((n/V)^3)$ and the total cost of $O(J n^3 / V^2)$, clarifying that choosing $V$ to grow with $n$ yields nearly linear effective complexity in sample size.
* **Robustness & Ablations:** We have better highlighted the ablation studies on the number of components $V$ and the experiments on **non-GMM conditioning sets** within the main text (Section 5), pointing readers to the detailed results in **Appendix A.1** which demonstrate robustness to model misspecification.
* **Limitations & Future Work:** In the Conclusion (Section 7), we explicitly acknowledged the computational limitation regarding large conditioning set dimensions ($|Z|$), which FastKCI shares with the standard KCI, and suggested combining our framework with methods addressing high-dimensional $Z$ as a direction for future work.
* **Presentation:** We incorporated minor presentational fixes and formatting improvements throughout the paper (e.g., font sizes in figures).

Overall, the reviews consistently acknowledge the strong motivation of our work and the effectiveness of FastKCI in substantially reducing computational costs while preserving the statistical power of the standard KCI test, which is a fundamental problem in causal discovery.

---

### Meta-Review · Area_Chair_fgu2 · 2025-12-07

**Summary:**

This paper introduces FastKCI, a scalable and parallelizable extension of the Kernel-based Conditional Independence (KCI) test to reduce its cubic computational cost. The method partitions data by fitting a Gaussian mixture model to the conditioning variables, performs independent KCI tests on each subset, and aggregates results via an importance-weighted sampling scheme. This divide-and-aggregate strategy preserves the statistical properties of the original KCI while enabling efficient computation on large datasets. Experiments on synthetic and real-world causal discovery tasks demonstrate that FastKCI maintains strong Type-I error control and competitive power compared to standard KCI, but with significantly reduced runtime.

The computational cost of KCI is cubic in the sample size. However, there are simple and effective techniques to accelerate kernel-based independence measures such as MMD and HSIC. A widely used approach is to randomly partition the data into blocks, compute HSIC or MMD within each block, and then average the results. This “block estimator” with permutation test strategy is straightforward and could similarly be applied to KCI. Since the proposed method appears closely related to this line of work, it is important to compare against such baselines.

[1] Large-scale kernel methods for independence testing. Statistics and Computing

Furthermore, while the Gaussian mixture–based partitioning is an interesting idea, the assumption of a GMM structure can be rather strong, and the theoretical guarantees remain limited. Therefore, I encourage the authors to further strengthen the theoretical analysis and include comparisons with block-based estimators before resubmitting to a future venue.

**Reviewer Concerns:**

1. Novelty of the approach.
2. The GMM assumption and its hyperparameter.
3. More practical experiments such as Sachs et al. (2005).
4. Other approximation techniques such as incomplete Cholesky decomposition, random Fourier features, or the Nyström approximation

Some of the concerns were well addressed. However, the authors did not clearly explain the advantages of their method over other approximation techniques, which should be compared against the proposed approach.

**Reviewer Scores:**

Based on the rebuttal, some reviewers might increase the score. However, baesd on the rebuttal, the negative reviewer will not increase the score.

---

### Decision · Program_Chairs · 2026-01-26

Reject